# Design and Characterization of Asymmetric Cell Structure of Auxetic Material for Predictable Directional Mechanical Response

**DOI:** 10.3390/ma15051841

**Published:** 2022-03-01

**Authors:** Rodrigo Valle, Gonzalo Pincheira, Víctor Tuninetti, Eduardo Fernandez, Esmeralda Uribe-Lam

**Affiliations:** 1Faculty of Engineering, University of Talca, Talca 3340000, Chile; r.vallefuentes@gmail.com; 2Department of Industrial Technologies, Faculty of Engineering, University of Talca, Talca 3340000, Chile; 3Department of Mechanical Engineering, Universidad de La Frontera, Francisco Salazar 01145, Temuco 4780000, Chile; victor.tuninetti@ufrontera.cl; 4Department of Aerospace and Mechanical Engineering, University of Liege, 4000 Liege, Belgium; efsanchez@uliege.be; 5Tecnológico de Monterrey, Escuela de Ingeniería y Ciencias, Querétaro 76130, Mexico; euribelam@tec.mx

**Keywords:** additive manufacturing, auxetic structures, cellular structures, mechanical characterization

## Abstract

A three-dimensional auxetic structure based on a known planar configuration including a design parameter producing asymmetry is proposed in this study. The auxetic cell is designed by topology analysis using classical Timoshenko beam theory in order to obtain the required orthotropic elastic properties. Samples of the structure are fabricated using the ABSplus fused filament technique and subsequently tested under quasi-static compression to statistically determine the Poisson’s ratio and Young’s modulus. The experimental results show good agreement with the topological analysis and reveal that the proposed structure can adequately provide different elastic properties in its three orthogonal directions. In addition, three point bending tests were carried out to determine the mechanical behavior of this cellular structure. The results show that this auxetic cell influences the macrostructure to exhibit different stiffness behavior in three working directions.

## 1. Introduction

In recent years, academic and industry research focusing on additive manufacturing (AM) has had an undoubted increase, mainly due to the possibility offered by this technology to fabricate components with complex geometries and less material waste compared to conventional manufacturing methods [1]. Another benefit of AM is the possibility of manufacturing cellular structures [2] or complex configurations generally formed by interconnected nodes and struts that maintain a periodic pattern [3,4]. These structures can present multiple attractive properties depending on the specific application, such as light weight, high specific stiffness, and high energy absorption capacity [5,6,7,8,9]. Sandwich panel fabrication is one of the most widely studied and reported in the literature [10,11,12,13,14,15]. Properties such as high impact energy absorption increase the use of cellular materials as protective structures [16,17,18]. The designs of these engineered structures are often inspired by nature, giving way to new cellular materials whose microstructure is present in natural biological configurations [19,20]. Examples are structures imitating the honeycomb cell, one of the most studied cases in lightweight applications, which are used in sandwich panels in the aerospace and packaging industry due to their high stiffness and low density [10,11,16,21,22].

The literature contains various studies of modeling and experimental characterization of cellular structures [23,24,25,26,27,28,29], showing that these microstructured materials can be designed to have a mechanical behavior which differs from that typically found in traditional materials. This is because their mechanical behaviors depend on both their chemical compositions and their subscale topology [3,4,10]. The topological design of cellular structures is therefore essential for customizing the mechanical properties of the material based on geometric parameters. For example, higher mechanical strength and rigidity are obtained with the increment of relative density. However, tensile-dominated cell designs may experience a higher Young’s modulus than bending-dominated designs [30,31] while maintaining constant density. In [32], an exhaustive review of the various morphologies of cell structures is presented, along with the superior properties that can be obtained, as well as their applications and challenges. One of the greatest interests in materials research is auxetic structures. Unlike conventional designs, they exhibit a negative Poisson’s ratio, i.e., they expand laterally when stressed and contract laterally when compressed. In [33], a review of the deformation mechanisms that allow auxetic behavior and the improved characteristics produced by a negative Poisson’s ratio is provided. This auxetic effect provides remarkable mechanical properties, such as low specific density, elevated shear moduli, high damping capacity, and toughness [34,35,36,37,38,39,40,41,42], essential properties requirements for the lightweight design of components and structures. Several design approaches for auxetic structures have been reported in the literature, from the pioneering analytical study shown by Almgren [43] of an isotropic auxetic structure with a Poisson’s ratio v=−1, through the work of Lakes in [44], where a foam structure with negative Poisson’s ratio is fabricated, to the works of Wojciechowski in [45,46] where a two-dimensional auxetic structure is studied through constant thermodynamic stress Monte Carlo simulations in order to determine its elastic properties. Subsequently, Gibson in [47] and Evans in [48] present an auxetic structure with re-entrant struts whose microstructure is modeled to exhibit transverse expansion under longitudinal loading. Finally, current studies reported where carbon fiber reinforced polymer additive manufacturing technology has been used using an interlocking assembly method in the fabrication of three-dimensional auxetic structures [49]. In addition, current research provides evidence and potential applications for the wood industry, specifically in furniture manufacturing, because plywood and bending processes consume significant time and resources, which could be considerably improved by implementing auxetic structures. For example, the potential application of sandwich panel manufacturing with auxiliary cores, enabling the reuse of materials [50,51,52]. The development of these structures allows the application of these auxiliary panels in a wide range of everyday products, such as doors and furniture [53], with improved impact energy absorption capacity and overall stiffness.

Depending on the required design application, the elastic properties of auxetic cells could be tailored as proposed in [54,55] where the design is studied through unit cell models using Timoshenko’s beam theory to manipulate Poisson’s ratio, Young’s modulus, and the elastic limit of the structure. This model shows great success in predicting the elastic behavior of this type of structure. Therefore, in [30], through theoretical analyses, numerical simulations, and experiments, different auxetic configurations are systematically analyzed to determine Poisson’s ratio and Young’s modulus.

In this way, there are analytical and experimental characterization studies to explore the mechanical properties of auxetic cellular structures; however, most of the structures are designed with homogeneous mechanical properties in their three directions of load. Therefore, they could only experience an orthotropic mechanical response by modifying their macroscale geometry. This requires a much more complex analysis to determine the effective mechanical behavior of the structure. This study focuses the theoretical and experimental analysis required to develop an auxetic structure with directional elastic behavior by using the re-entrant structs. Using classical Timoshenko beam theory, the deflection of struts is modeled to adapt Poisson’s ratios and Young’s moduli, thus creating a new orthotropic structure.

The paper presents in Section 2 the classical beam theory adapted for three-dimensional topological design with orthotropic response, including the experimental methods for mechanical characterization of the macrostructures. Section 3 provides the experimental results correlated with the model, and finally, Section 4 estates the main findings and the need for future work.

## 2. Materials and Methods

### 2.1. Orthotropic Mechanical Model of the Cell

Based on the models developed in [54,55,56,57], this analysis uses the same 2D auxetic cell with re-entrant struts as the base structure to design an asymmetric 3D cell. As a differentiator of similar structures studied in [58,59], this study includes a new design parameter ∝ that allows the manipulation of the asymmetry of the cell  (0<∝<0.5), as shown in Figure 1b highlighted in red. This asymmetry allows the development of a 3D auxetic cell that can present orthotropic mechanical behavior. To model the mechanical behavior of this cell in its three main directions, Timoshenko beam theory is applied. This model is developed based on 5 parameters of the cell: the vertical length H, the length of the re-entrant struts L, the re-entrant angle θ, the thickness of the cross-section t, and the new parameter ∝ that allows to manipulate the position in which the 2D cells intersect. It is worth mentioning that the asymmetry cell does not prevent connectivity between neighboring cells to form a macrostructure, as shown in Figure 1c.

As is known, the mechanical behavior of this type of structure is dominated by the bending of its re-entrant elements. As Timoshenko’s beam theory has been shown to successfully model the mechanical behavior of this type of cell [30], the simplified method is applied (Figure 2) that shows the connection of two rigidly joined struts, experiencing the superposition of three components of deformation Figure 2a:

Axial deformation of the vertical elements $\Delta y_{I}$;

Deflection of the re-entrant element $\Delta y_{II}$;

Shear induced by deflection of the re-entrant element Δx**,**
Δz as appropriate for compression in the x or z direction, respectively.

Furthermore, experimental analysis demonstrated that the re-entrant angle θ undergoes a negligible variation during the deformation of the structure. This allows the consideration of rigid joints and the study of re-entrant struts as a cantilever beam [56]. Therefore, the deformation components can be computed in terms of the maximum angle of deflection $\theta_{II}$, according to the tributary area that each element supports. From Timoshenko’s theory, this angle is defined by:(1)$$\theta_{II}=\frac{ML}{6E_{s}I}+\frac{6P}{5G_{s}A}$$
where $E_{s}$ and $G_{s}$ are the Young’s and shear moduli, respectively. A is the cross-sectional area, I is the moment of inertia of the cross-section, M corresponds to the bending moment, and P is the applied load. In this study, as the cross-sectional thickness t of the elements is much smaller than the length L, the structure is simplified as an Euler–Bernoulli beam. In addition, the fact that $1/t^{2}\ll L^{2}/t^{2}$, a negligible compression of the vertical strut can be considered according to [57]. For more details, the reader is referred to [56].

In the following, three load cases acting in the principal directions of the cell are analyzed in order to obtain the mechanical properties of the cell.

#### 2.1.1. *x* Axis Compression

When applying compressive loads to the structure in the x-direction, the stresses are transmitted through cell 1 shown in Figure 1a. As the edges of each re-entrant element 1 and 2 are shared by two adjacent cells, the compressive stress is distributed in the section of two neighboring cells. According to the tributary area supported by each strut type i={1, 2}, the compression force acting on the element is $F_{i}$, as shown in Figure 2b. For this case, as cell 1 is symmetric, the compression force transmitted by each strut is the same:(2)$$F_{1}=F_{2}=\sigma \;L\;sin\;\theta \;H$$
where σ is the compression stress. With the compressive load, it is possible to compute the internal reactions of each strut (denoted as T, P, and M in Figure 2c) to later calculate their deformations. As cell 1 (b) is symmetric with respect to the *y* axis, struts 1 and 2 will exhibit equal deformations. Therefore, Poisson’s ratio $v_{yx}$ can be estimated based on the geometry, as:(3)$$v_{yx}=-\frac{\varepsilon_{y}}{\varepsilon_{x}}=-\frac{\left(\Delta y_{I}+\Delta y_{II}\right)L\;sin\;\theta \;\;}{\Delta x\left(H-L\;cos\;\theta \;\right)}=-\frac{sin^{2}\;\theta \;}{cos\;\theta \;\left(\frac{H}{L}-cos\;\theta \;\right)\;}$$
where $\varepsilon_{x}$ and $\varepsilon_{y}$ correspond to the strains in the x and y direction, respectively. Similarly, Young’s modulus $E_{x}$ can be estimated, as:(4)$$E_{x}=\frac{\sigma }{\varepsilon_{x}}=\frac{\sigma L\;sin\;\theta \;}{\Delta x}=\frac{2t^{4}}{L^{3}Hcos^{2}\theta \;}E_{s}$$

#### 2.1.2. *z* Axis Compression

When applying compressive loads to the structure in the z-direction, cell 2 shown in Figure 1b becomes the support structure. According to the tributary area supported by each strut type i={3, 4} the compression force $F_{i}$ transmitted to each strut is the same:(5)$$F_{3}=F_{4}=\sigma L\;sin\;\theta H\;$$
where σ is the compression stress. The internal reactions T, P, and M of each strut are shown in Figure 2c. Therefore, according to the Euler–Bernoulli theory the Poisson’s ratio $v_{yz}$ can be estimated for each strut i, as:(6)$${\left(v_{yz}\right)}_{i}=-\frac{sin^{2}\;\theta^{*}\;}{cos\;\theta^{*}\;\left(\frac{H}{L^{*}}-cos\;\theta^{*}\;\right)\;}$$
where the geometric parameters $\theta^{*}=\left\{\theta^{\prime },\;\theta^{\prime \prime }\right\}$ and $L^{*}=\left\{L^{\prime },\;L^{\prime \prime }\right\}$ correspond to strut type 3 or 4, respectively. Similarly, the Young’s modulus $E_{z}$ can be estimated for each strut i based on its geometry, as:(7)$${\left(E_{z}\right)}_{i}=\frac{\sigma t^{2}sin\;\theta^{*}}{F_{i}{\left(L^{*}\right)}^{2}\;cos^{2}\theta^{*}}E_{s}$$

In this way, the elastic behavior of the cell under compression on the z axis can be represented by the Poisson ratio ${\bar{v}}_{yz}$ and a Young’s modulus ${\bar{E}}_{z}$ averaged between struts 2 and 3.

#### 2.1.3. *y* Axis Compression

When applying compressive loads to the structure in the y direction, cells 1 and 2 form the supporting structure. Unlike other auxetic structures, in this case, there will be an asymmetric distribution of the load in the re-entrant support elements. As this asymmetry is dependent on the design parameter ∝, the compressive load acting on each re-entrant element is also dependent. Consequently, according to the contributing area supported by each strut i, where i={1, 2, 3, 4}, the compressive load $F_{i}$ transmitted to each strut is:(8)$$F_{1}=F_{2}=3\sigma L^{2}sin^{2}\theta \;\left(\frac{1}{8}+\frac{\propto }{2}-\frac{\propto^{2}}{2}\right)$$
(9)$$F_{3}=3\sigma L^{2}sin^{2}\theta \;\left(1-2\propto +\propto^{2}\right)$$
(10)$$F_{4}=3\sigma L^{2}sin^{2}\theta \;\left(\propto -\frac{1}{4}\right)$$
where σ is the compression stress. The internal reactions T, P, and M of each strut are shown in Figure 2b. Similarly, the Poisson ratios $v_{xy}$ and $v_{yz}$ can be estimated by analyzing the deformations for each strut, such as:(11)$$v_{xy}=-\frac{cos\;\theta \;\left(\frac{H}{L}-cos\;\theta \;\right)}{sin^{2}\;\theta }$$
(12)$${\left(v_{yz}\right)}_{i}=-\frac{cos\;\theta^{*}\;\left(\frac{H}{L^{*}}-cos\;\theta^{*}\;\right)}{sin^{2}\;\theta^{*}\;}$$
where the geometric parameters $\theta^{*}=\left\{\theta ,\;\theta^{\prime },\;\theta^{\prime \prime }\right\}$ and $L^{*}=\left\{L,\;L^{\prime },\;L^{\prime \prime }\right\}$ correspond to strut 3 or 4, respectively. Finally Young’s modulus $E_{y}$ can be determined for each strut i based on its geometry, as:(13)$${\left(E_{y}\right)}_{i}=\frac{\sigma t^{4}\left(\frac{H}{L^{*}}-cos\theta^{*}\right)}{F_{i}{\left(L^{*}\right)}^{2}sin^{2}\theta^{*}}E_{s}$$

In this way, the compressive elastic response in the y axis of the cell can be represented by the average value of the Young’s moduli of struts 1, 2, 3, and 4.

#### 2.1.4. Poisson’s Ratio

Using the design parameters H, L, t, θ, and ∝*,* it is possible to completely describe the mechanical behavior of this auxetic structure in its three working directions. By analyzing the deflections of each re-entrant element can be established an average value for the Poisson’s ratio. Figure 3 shows the Poisson’s ratio as a function of the re-entrant angle θ and the distance factor ∝.

This graphic permits a rapid parameter selection to obtain an adequate mechanical behavior under shear stresses, as it is known that structures of this type contract transversely when subjected to compression. Consequently, an additional shear stress is generated in the cross-section plane. It has been shown experimentally that the shear modulus increases as the Poisson’s ratio becomes increasingly negative [57].

#### 2.1.5. Young’s Modulus

Similarly, the average Young’s modulus can be plotted in its three orthogonal directions as a function of the re-entrant angle θ and the distance factor ∝. As Equations (4), (7), and (13) depend on the Young’s modulus E of the material, the normalized modulus ${\bar{E}}_{i}/E_{s}$ can be expressed for each direction, as shown in Figure 4.

It can be seen from the curves shown in Figure 4 that the distance factor ∝ does not affect the Young’s modulus $E_{x}$, while $E_{z}$ is slightly dependent on ∝. This result is explained by the strong dependence of the moduli on the design parameters $\theta^{*}$ and $L^{*}$ derived from the asymmetry of the cell. This also produces a difference between ${\underline{E}}_{x}$ and ${\underline{E}}_{z}$, resulting in an orthotropic auxetic structure.

### 2.2. Experimental Procedure

Experimental samples were constructed using a 3D printer Stratasys uPrint SE, equipped with Fused Deposition Modeling (FDM) technology. The raw filament material is ABSplus. Due to the intrinsic anisotropy of the FDM process, the Young’s modulus of the raw material, denoted as $E_{s}$, was characterized in three different configurations to obtain the components ${\left(E_{s}\right)}_{x}$, ${\left(E_{s}\right)}_{y}$, and ${\left(E_{s}\right)}_{z}$. ${\left(E_{s}\right)}_{y}$ represents the Young’s modulus in the building direction and is measured using the dog bone specimen $D_{y}$, shown in Table 1. ${\left(E_{s}\right)}_{x}$ and ${\left(E_{s}\right)}_{z}$ are the Young’s modulus in the x and z directions, and are measured using the dog bone specimens $D_{x}$ and $D_{z}$, respectively. The obtained Young’s modulus is shown in Table 1. Note that ${\left(E_{s}\right)}_{x}={\left(E_{s}\right)}_{z}$, which indicates isotropy with respect to the xz plane. To ensure representativeness of the raw material and to reduce the isotropic or orthotropic effect of the FDM process, the specimens fabricated to measure the properties of the proposed cell are oriented such that the xz plane of the cell corresponds to the baseplate and the y axis to the printing direction. Under this configuration, the ∝ parameter is leading to the anisotropy of the cells.

Three groups of 32 unit cell macrostructures (4×4×2) were fabricated using different values for the re-entrant angle θ={50°; 60°; 70°}. Furthermore, three samples were manufactured within each group with different values for the parameter ∝={0.27; 0.33; 0.4}. The dimensions and geometric parameters for each macrostructure are shown in Figure 5.

To validate the design of the structure, quasi-static compression tests were performed on a Zwick Roell Z005 test system (Zwick/Roell GmbH, Ulm, Germany) with a maximum load capacity of 5 KN, under a constant cross-head speed of 1 mm/min. Compressive loadings were applied to the samples in three principal directions to statistically obtain Young’s moduli and Poisson’s ratios (Figure 6). Each sample was subjected to compression in the x, y, and z directions within the elastic range. During the experiment, the test machine automatically recorded the value of load and displacement along the compression direction. The Poisson’s ratio for each sample was obtained by stopping the machine and measuring the sample’s dimensions in the transverse directions using a dial gauge. To improve the precision of the measurement and reduce the experimental error, the displacement was measured three times to compute the mean value.

To evaluate the mechanical behavior that this cell can impart to a macrostructure subjected to bending, three groups of macrostructures of 54 unit cells (3×3×6) were manufactured using different values for the re-entrant angle $\theta =\left\{50^{{}^{\circ}};\;60^{{}^{\circ}};\;70^{{}^{\circ}}\right\}$. Within each group, three samples were made with different values for the parameter ∝={0.27; 0.33; 0.4}, as shown in Figure 7.

Quasi-static bending tests were then carried out at a constant speed of 1 mm/min. Each of the samples was tested in the x and z directions within the elastic range. In the latter case, the sample receives the load in the positive direction (+z) and then the sample is rotated 180° to receive the load in the negative direction (−z), as shown in Figure 8. A different mechanical behavior is expected to occur due to the asymmetry of the structure in this direction. During the experiment, the test machine automatically recorded the value of the force and the vertical displacement along the bending direction. On the other hand, as shown in Figure 7, the length of the macrostructures varies as a function of the re-entrant angle. All the beams are six unit cells long, so to establish a representative comparison between all the macrostructures, bending tests were carried out with a distance between supports equivalent to the length of four unit cells.

The experimental results from compressive tests were correlated with the properties resulting from the theoretical mechanical model provided in Section 2.1. As a limit of the load was imposed to avoid causing irreversible deformations in the specimen structures, their maximum compressive strength was not obtained. The bending results were used to evaluate the bending stiffness behavior for each direction and to evaluate possible applications of the structure.

## 3. Results and Discussion

To validate the values obtained for the Poisson’s ratios and Young’s moduli of the structures, twenty seven quasi-static compression experiments were performed. Nine structures were fabricated with the parameters shown in Figure 5. Each sample was compressed in its three orthogonal directions. The results show a high consistency, which allows a valid analysis to calculate the mechanical properties. On the other hand, 27 three-point bending experiments were also performed. Nine beams with the same geometric parameters were manufactured to determine the stiffness that this asymmetric cell can transmit to the macrostructure. At all times, care was taken to end the experiments with very small strain values to avoid permanent damage. The experimental values are shown in Figure 9, Figure 10, Figure 11 and Figure 12.

### 3.1. Poisson’s Ratio

To determine the elastic behavior of this cell structure, nine combinations are achieved for the Poisson’s ratio: three groups for different values of the re-entrant angle θ and three groups for the new design parameter ∝. The theoretical and experimental results for Poisson’s ratio are shown in Figure 9. In general, the Poisson’s ratio curves as a function of the new design parameter ∝ show high accuracy with the experimental results. The theoretical model achieves capture the dependence of the Poisson’s ratio with the parameter ∝. As the structure experiences a homogeneous deformation under compression in the x direction, the Poisson’s ratio does not vary as a function of ∝, but slightly decreases as a function of the re-entrant angle. On the other hand, due to the asymmetry of cell 2 shown in Figure 1b, the Poisson’s ratio yz becomes more negative the smaller the value of ∝, and it decreases slightly the smaller the re-entrant angle, which helps to validate the proposed design. However, in some cases, the disagreement can be as high as 20%, such as structures designed with a 50° re-entrant angle. However, it should be noted that the theoretical analysis is based on the assumptions that the structure has an infinite pattern of cells in the three orthogonal directions and considering only the bending deformation through Timoshenko’s theory. Furthermore, the analysis only considers the bending deformation through Timoshenko’s theory, while the measurements on the samples that have a finite number of cells and the samples experience multiple deformations such as axial and shear. On the other hand, the application of the simplified model becomes more complex to apply to this cell due to the asymmetry of cell 2 shown in Figure 1b. Another possible cause of the differences between the theoretical predictions and the experiments is the variability in the size of the strut and the quality of the surface produced by the manufacturing process, particularly considering that structures of this type with re-entrant struts inevitably have defects such as the well-known step effect produced by additive processes [60]. On the other hand, the Poisson’s ratio obtained through the Euler–Bernoulli approximation only depends on the geometry of the structure and does not consider the properties of the material. According to research carried out in [61,62], auxetic structures that contradict the theoretical behavior are studied by experiencing a positive Poisson’s ratio. In this way, these results show that the properties of these metamaterials not only depend on their microstructure and that the manufacturing material can be crucial. Therefore, it is a challenge to model with great precision the transverse displacement of this type of structures under compressive loads.

As reported in [55], the experimental Poisson’s ratio results exhibited by the cell differ from the theoretical values; nevertheless, the trend of the auxetic behavior of the experimental cell agrees well with theoretical predictions, as there is a difference between the Poisson’s ratio $v_{yx}<v_{yz}$ just as predicted with the analytical model. On the other hand, the object of the approach used here, unlike other studies such as [30], is to design a cell with different elastic properties in its three orthogonal directions; hence the importance of the Poisson’s ratio being  v ≠1, in order to avoid equality between the transverse deformation and the longitudinal deformation under each compression direction. According to [54], it is possible to achieve control of Young’s modulus values over a wide range by controlling Poisson’s ratio values. In this way, the behavior of the structure can be quickly manipulated by the design parameters to obtain the desired elastic behavior. It is therefore of great importance to validate Equations (3) and (6) to demonstrate the orthotropic behavior of this auxetic cell. It is important to highlight that the magnitude of the Poisson’s ratio can be adapted by manipulating the geometric parameters and also through the hierarchical configuration between the primary and secondary structure (Figure 1a,b). The results of this research have been carried out with specific parameters and the dependence of the hierarchical 3D configuration on the elastic properties is intensively investigated in our future studies.

### 3.2. Young’s Modulus

Figure 10 shows the experimental results in the macrostructures subjected to compression, for three values of $\theta =\left\{50^{{}^{\circ}};\;60^{{}^{\circ}};\;70^{{}^{\circ}}\right\}$ and for three values of ∝={0.27; 0.33; 0.4}. The experimental results show that there are differences in the elastic modulus for each compression direction in all cases. It can be seen that the elastic modulus in the z direction decreases in magnitude as the parameter ∝ diminishes. Furthermore, as expected, the parameter ∝ has no influence on the module in the x direction, as this direction is conditioned by the homogeneous deformation of cell 1 shown in Figure 1a.

The correlation of theoretical results with the experimental Young’s moduli (x, y, and z) for three groups with different re-entrant angles θ and three groups for the new design parameter ∝ is illustrated in Figure 11. High consistency in results is achieved between the experiment and the model, which helps to validate Equations (4), (7), and (13), as the trend of the mechanical behavior in the x and z directions agree well with the theoretical predictions. As the value of ∝ diminishes, the module $E_{z}$ decreases with respect to $E_{x}$; and the smaller the value of the re-entrant angle θ, the greater the difference. The AM process introduces intrinsic anisotropy in the structure due to the layer by layer process and generates a dependency of the structure with the manufacturing orientation. However, the orthotropic behavior of this cell is clearly induced by ∝, as all the samples were printed on the xz plane. This allows the geometric design of the cell to be selected quickly according to the proposed application. However, in all cases, Young’s modulus $E_{y\;\left(\propto =0.27\right)}$ differs widely from the model. This is because, mathematically, the model tends to infinity around ∝ ≈0.25 for Young’s modulus $E_{y}$. Therefore, it was to be expected that for ∝=0.27 the experimental value would be below the theoretical value. According to [55], the poor correlation between the model and the experiments may be due to the inaccuracy derived from the effective length $L^{*}\;$ of the re-entrant struts. Furthermore, the theoretical Young’s modulus depends on the modulus of the material, which has been experimentally determined for different printing planes. In addition, in the design, the macrostructure is considered as a homogeneous continuous, which is then analyzed with this traditional technique for the analysis of beams. However, this approach faces significant challenges for boundary conditions. Considering that structural asymmetry results in different force and moment components for each support strut.

The theoretical development of this study is a good approximation that allows us to design this auxetic structure with orthotropic mechanical behavior. Due to the mathematical simplicity demonstrated in this work, the unit cell design could easily be incorporated into advanced structural design for real structures. Furthermore, this design approach could be applied in future work to explore the mechanical behavior of other hierarchical configurations between cell 1 and 2 (Figure 1a,b).

### 3.3. Flexural Stiffness

It is important to note that the approach developed in [54,55,56,57] has been applied to model the asymmetric structure. However, unlike these publications, in this work, an orthotropic auxetic cell is designed. The results obtained are in good agreement with the experiments and it is shown that the proposed design transmits an orthotropic behavior to the macrostructure under compressive stresses. Therefore, in order to explore in depth the mechanical behavior that this cell can achieve, Figure 12 shows the results obtained experimentally from the macrostructures subjected to bending, for three values of θ={50°; 60°; 70°} and for three values of ∝={0.27; 0.33; 0.4}.

A difference in the stiffness behavior under load force in the x and ±z directions is demonstrated in Figure 12. The product of the asymmetry of cell 2 shown in Figure 1b generates a difference in the stiffness behavior under load force in the +z and −z directions. Furthermore, this difference in stiffness behavior increases as the design parameter ∝ and the re-entrant angle θ diminish. In this way, this orthotropic cell can also impart to a macrostructure a different stiffness behavior in three directions under bending loads. This would allow the application of this cell in structures that require differentiated mechanical behavior according to the direction of work, such as a prosthetic ankle. Generally, this type of structure is made from polymers which are not normally rigid enough for structural applications. However, it is hoped that stiffness can be significantly increased by a good selection of geometric parameters.

## 4. Conclusions

This study presents the design of an orthotropic auxetic cell structure. Through topology analysis and using classical Timoshenko’s beam theory, for adapting the Young’s modulus and the Poisson’s ratio of the structure, five geometric parameters were defined. The mechanical properties of the macrostructure were measured in three orthogonal directions by compression experiments.

The Timoshenko’s beam theory can be applied for predicting the elastic properties of the studied re-entrant strut structure with good accuracy, allowing us to consider the Timoshenko deflection model as a quick design approach for the use of the proposed orthotropic cell in several future applications.

The nine macrostructures manufactured with the FDM system and ABSplus considering different values of the distance factor ∝ and the re-entrant angle θ were loaded under monotonic compression in their three orthogonal directions to determine the elastic mechanical behavior.

Young’s modulus and theoretical Poisson’s ratio presents high concordance with the values obtained from the experiments, demonstrating that the method can be used as a design tool to adapt the elastic moduli of the macrostructure. The results demonstrated that the proposed design for this auxetic cell can achieve a macrostructure with global orthotropic mechanical performance easily manipulated through the design parameter ∝. In addition, using the same combinations of geometric parameters, nine macrostructures were fabricated for bending experiments. The results obtained demonstrated that this asymmetric cell can transmit to the macrostructure a different stiffness behavior in three directions, generally required for design of special structures with different mechanical behavior depending on the direction in which the loads are applied. This cell greatly expands the potential of auxetic structures for engineering applications, as the results obtained in this work are applicable in real structures such as, for instance, an ankle prosthesis. In addition, it would allow new opportunities for academic research and for industry as these reported results would also offer a novel way to explore the energy absorption and impact mechanics behavior in the auxetic family. Ongoing research work is in progress to obtain higher accuracy predictions, particularly considering the inherent anisotropic issues associated with the current manufacturing FDM process, for which the validation of the model with SLS technology is fundamental. 

## Figures and Tables

**Figure 1 materials-15-01841-f001:**
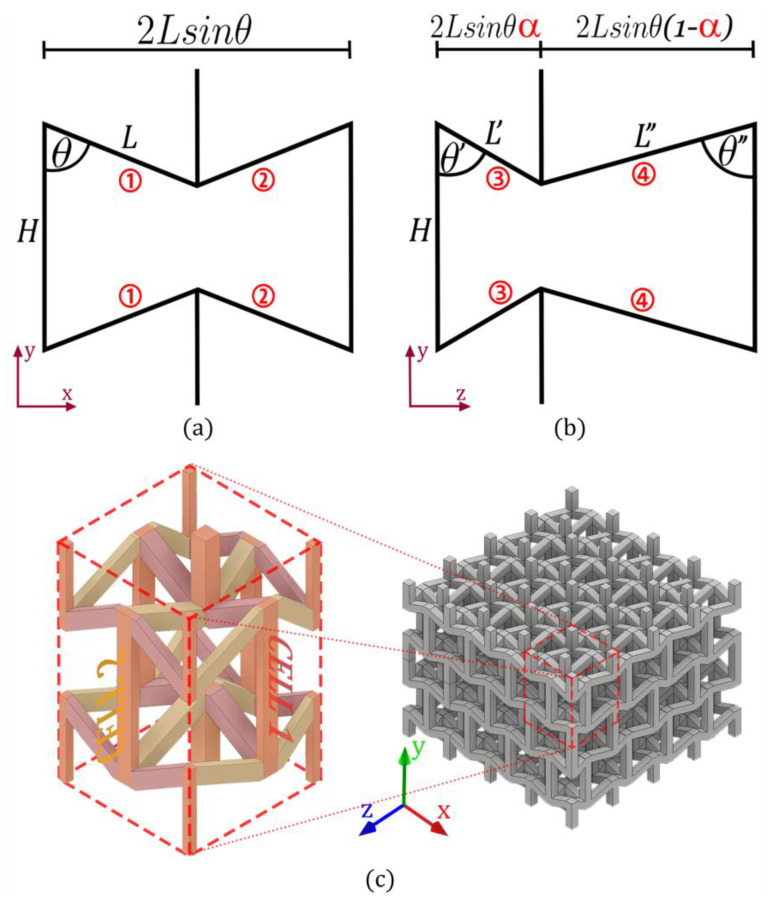
Orthotropic auxetic structure design with re-entrant struts. Where (**a**) corresponds to a 2D re-entrant cell (cell 1), which intersects at the re-entrant vertex with an asymmetric 2D re-entrant cell (cell 2) (**b**) to form an asymmetric 3D auxetic cell. This unit cell can be replicated to form a macrostructure ensuring connectivity between cells (**c**).

**Figure 2 materials-15-01841-f002:**
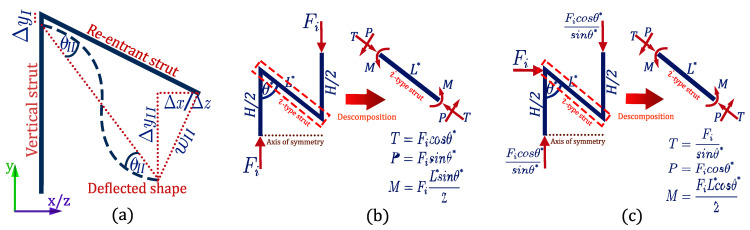
Free body diagram of the auxetic structure. Where (**a**) illustrates the deformations of each re-entrant strut, (**b**) the internal loads under compression on axis y, while (**c**) illustrates the distribution of the internal loads under compression on the x and z axis.

**Figure 3 materials-15-01841-f003:**
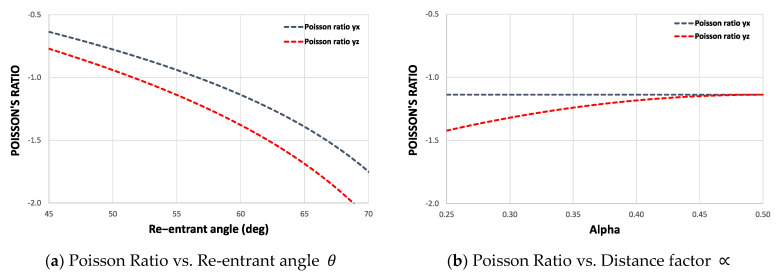
Average Poisson’s ratio for different design parameters. Considering H=10 mm, L=5.5 mm, and t=1.5 mm.

**Figure 4 materials-15-01841-f004:**
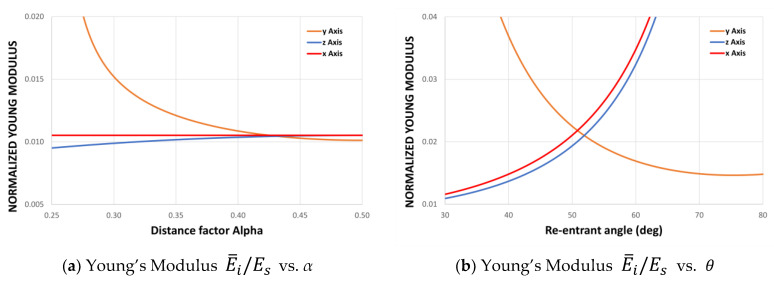
Normalized Young’s modulus ${\bar{E}}_{i}/E_{s}$ for different values of the re-entrant angle θ and for different values of the design parameter ∝. Considering H=10 mm, L=5.5 mm, and t=1.5 mm.

**Figure 5 materials-15-01841-f005:**
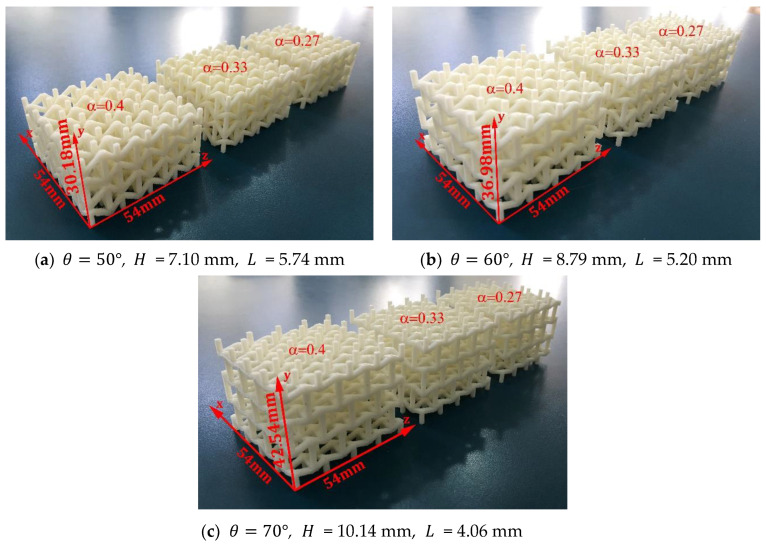
Macrostructures built up of 32 unit cells (4×4×2) from fused filament fabrication, considering t = 1.50 mm.

**Figure 6 materials-15-01841-f006:**
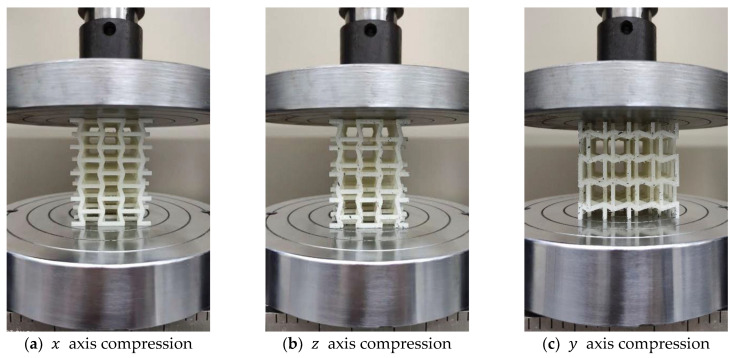
Compression experiments on each proposed macrostructure.

**Figure 7 materials-15-01841-f007:**
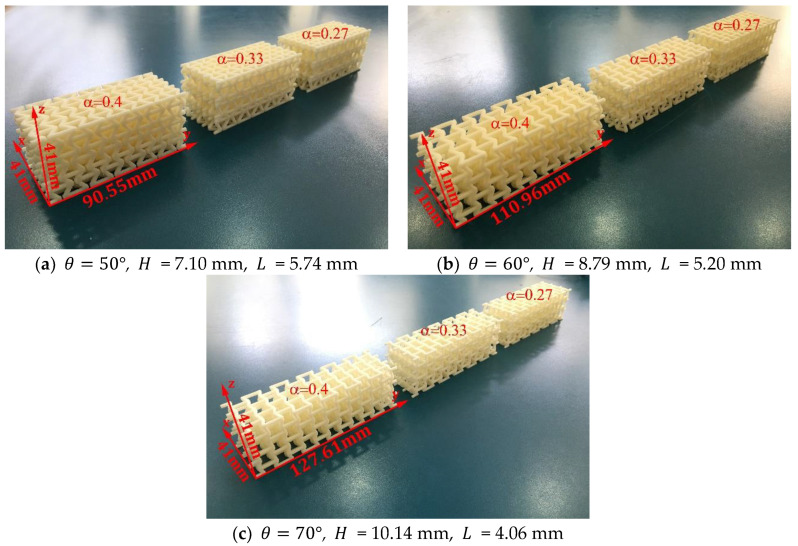
Beams made up of 54 unit cells (3×3×6) manufactured through fused filament fabrication, considering t = 1.50 mm.

**Figure 8 materials-15-01841-f008:**
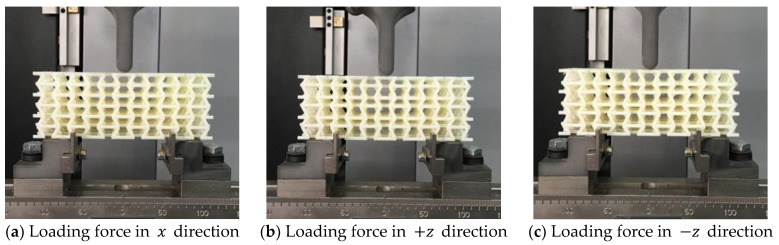
Bending experiments of the proposed macrostructure in the x, +z, and −z direction.

**Figure 9 materials-15-01841-f009:**
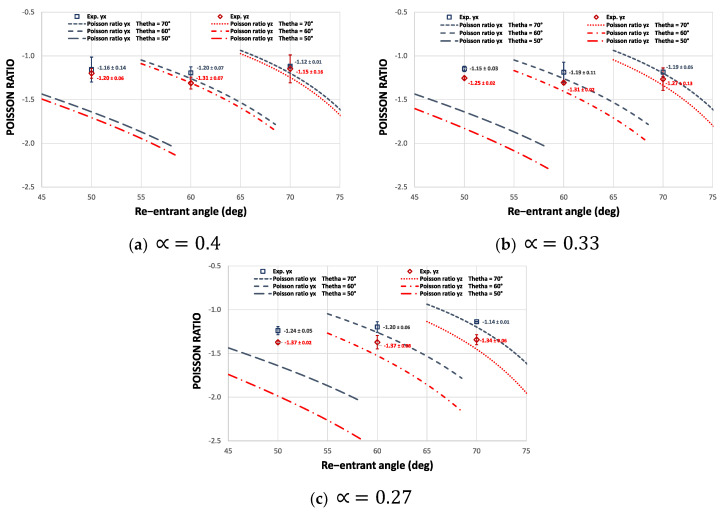
Comparison of the theoretical Poisson’s ratio vs. the experimental Poisson’s ratio for each macrostructure.

**Figure 10 materials-15-01841-f010:**
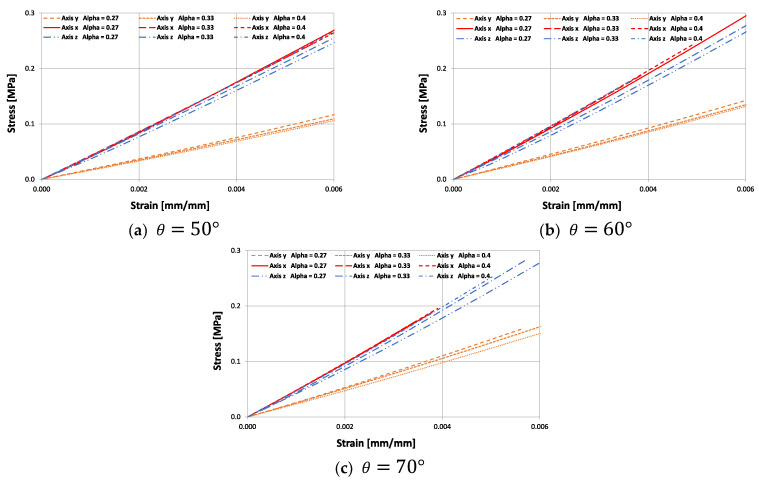
Stress vs. strain curves obtained experimentally under quasi-static compression for the 3D cell in its three orthogonal directions for different values of ∝={0.27; 0.33; 0.4}.

**Figure 11 materials-15-01841-f011:**
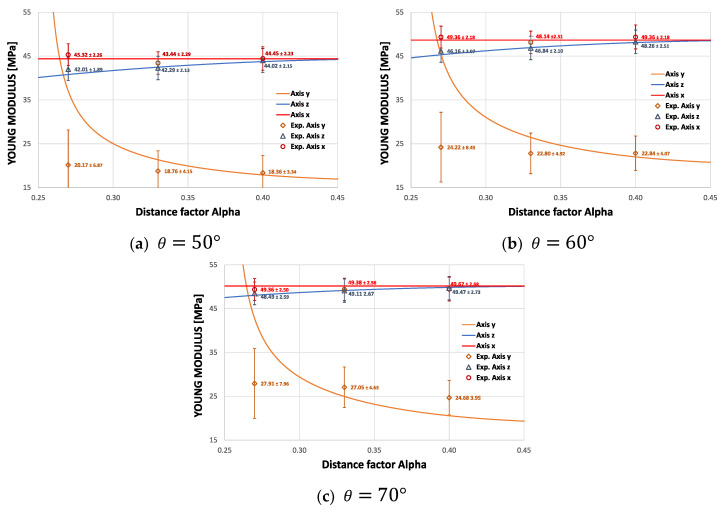
Experimental and theoretical Young’s moduli ${\underline{E}}_{i}$ for each value of ∝={0.27; 0.33; 0.4}.

**Figure 12 materials-15-01841-f012:**
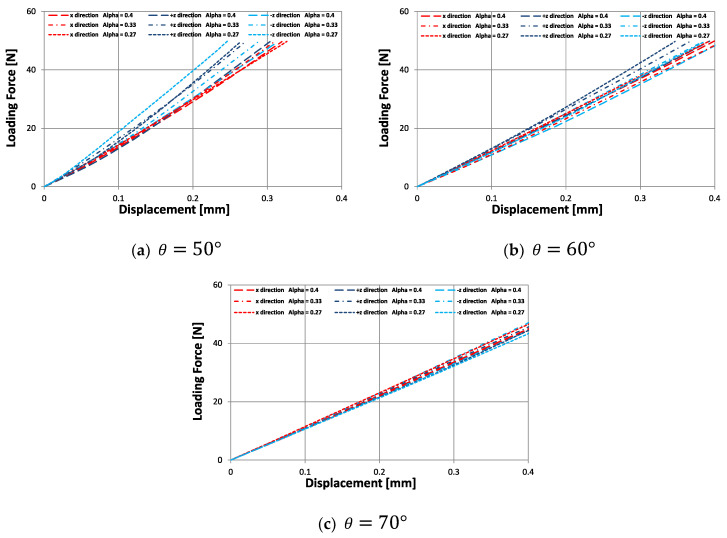
Bending tests performed on the macrostructures of 3×3×6 cells, under load force in the x, +z, and −z directions.

**Table 1 materials-15-01841-t001:** Measured Young’s Modulus of the raw material (ABSplus) obtained by traction tests using dog bone specimens. The baseplate and printing direction correspond to the *xz* plane and *y* axis, respectively.

*ABSplus*	
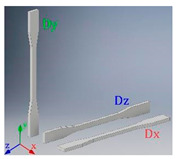	${\left(E_{s}\right)}_{y}=903.3\;\pm 167\;\mathrm{MPa}$
	${\left(E_{s}\right)}_{x}=1927.2\pm 86\;\mathrm{MPa}$
	${\left(E_{s}\right)}_{z}=1927.2\pm 86\;\mathrm{MPa}$
	$\rho =0.98\;g/{\mathrm{cm}}^{3}$

## Data Availability

Data sharing is not applicable to this article.

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
