# Peer review of "Design and Characterization of Asymmetric Cell Structure of Auxetic Material for Predictable Directional Mechanical Response"

_materials, 2022, doi:10.3390/ma15051841_

Round 1

Reviewer 1 Report

Response letter

The submitted manuscript entitled “Design and characterization of asymmetric cell structure of auxetic material with directional mechanical response” provide the report on the design and the characterisation of novel auxetic material.

The article’s topic is not novel, but is addressed in an appropriate, partly non-consistent and non-thorough way. The manuscript is interesting, but major revision should be done before publishing.

General comments:

  1. The structure presented in this paper is not new. 3D re-entrant auxetic structures were widely studied in previous works in literature. The only difference is the introduction of a new design parameter ∝ that allows the manipulation of the asymmetry of the cell. But when changing this, the structure becomes very similar to the auxetic structures called inverted tetrapods, already studied before:

Schwerdtfeger J, Heinl P, Singer RF, Körner C. Selective Electron Beam Melting : A New Way to Auxetic Cellular Structures. Mater Sci Technol 2009:724–9.

Novak N, Vesenjak M, Ren Z. Computational Simulation and Optimization of Functionally Graded Auxetic Structures Made From Inverted Tetrapods. Phys Status Solidi B 2017;254. doi:10.1002/pssb.201600753.

Therefore, please avoid many statements about »new structure« and cite the existing research in this field.

Comments that should be addressed in manuscript before publishing:

  1. Figure 3: provide graphs of better quality, i.e. position of z-axis label on figure a.)
  2. Provide more info about the used testing machine.
  3. Do the authors take into account the relaxation of the material, when stopping the loading procedure in order to measure the lateral displacement?
  4. The measuring of the Poisson’s ratio can be easily done with freely available DIC tools like GOM Correlate. The reviewer’s opinion is that the measurement of the transverse displacement in the range of tenths of a millimetre cannot be accurate enough – at least from the description provided in the paper.
  5. Figure 9, the Poisson’s ratio in cellular structures is strain dependent – at which strains are the values given?

Author Response

Dear Reviewer

We appreciate the opportunity offered for improving our manuscript and resubmit it to the journal Materials for a second round of revisions. Regarding the comments of reviewer 1, we have improved all the recommendations in order to be considered for publication. Additionally, Reviewer 1 also recommended the use of free software for image analysis (GOM Correlate), This has been included.

To conclude, all specific reviewers' comments were considered and included in the review. Please note that modified and inserted texts are clearly highlighted, because we have used the “track of changes” option in Microsoft Word (materials-1452048_track_changes). Additionally, we have added a final version of the work with all the changes accepted (materials-1452048_final). Finally, together with this letter, you will find two letters addressed to reviewers addressing their comments. We hope the changes we made in our paper will make it suitable to be published in your journal.

Thank you very much for your consideration.

Yours sincerely,

The authors

Reviewer 2 Report

The article is devoted to the analysis of the mechanical properties of structures built on the basis of asymmetric reentrant honeycomb cells.
The article does not bring much new in the topic of examining the mechanical properties of auxetic structures and I do not think that it should be published in its present form. Below is a list of some issues authors should consider when improving their article
1.    Mechanical analysis of a typical auxetic reentrant honeycomb structure is given in the classic book by Lorna J. Gibson, Michael F. Ashby, "Cellular Solids: Structure and Properties," Cambridge University Press, May 1, 1997
2.    The analysis of the basic loading conditions presented in Figure 2 makes sense in the case of a periodic structure. However, the article concerns a structure composed of asymmetric cells, which means that the structure indicated in Figure 2 is not an elementary cell of periodic material. 
Compression of the metamaterial composed of asymmetric cells will result in the individual cells not being subject to only axial compression, as shown in Figure 2b. Asymmetric cells cannot be divided into parts as if they were symmetrical and considered separately.
3.    When analyzing the plots (Fig. 9), it can be concluded that the Poisson ratios obtained in the experiments for different values of the alpha parameter do not change and are closer to each other than the plots of the analytical values.
Similarly, looking at graphs 10 and 12 - it shows that the asymmetry parameter \alpha has no effect on the structure, which seems quite surprising.
4.    In turn, in Figure 10 it can be seen that the experimental results coincide with the analytical results only in the case where the values of the \alpha parameter remain almost constant (x- and z-direction), while in the case of the y-direction they are very different from each other.
5.    The formulas are poorly edited, it is not known what is a function, what is a variable, and what is an argument of a function, e.g. in formula (2) it looks as if \ theta is a separate variable but I guess it should be an argument of a sine. Moreover, in formula 3 and the following it is difficult to interpret the way of writing the trigonometric functions at all
6.    All the values obtained in the experiments should be given directly in numerical form, and not only marked on the graph, as well as the values of uncertainty

Author Response

Dear Reviewer

We appreciate the opportunity offered for improving our manuscript and resubmit it to the journal Materials for a second round of revisions. We have addressed all the points raised by the reviewers. Regarding the comment of reviewer 2, authors have tried to answer the reviewer's questions in the best way possible with specific information. Additionally, we have included all the recommendations.

To conclude, all specific reviewers' comments were considered and included in the review. Please note that modified and inserted texts are clearly highlighted, because we have used the “track of changes” option in Microsoft Word (materials-1452048_track_changes). Additionally, we have added a final version of the work with all the changes accepted (materials-1452048_final). Finally, together with this letter, you will find two letters addressed to reviewers addressing their comments. We hope the changes we made in our paper will make it suitable to be published in your journal.

Thank you very much for your consideration.

Yours sincerely,

The authors

Reviewer 3 Report

This paper attempts to investigate the design and characterization of auxetic structures which has orthotropic mechanical properties. This study lacks novelty since several authors have already investigated auxetic structure mechanical properties and it is well known that these structures have a negative Poisson’s ratio and behaves like an asymmetric orthotropic material. Therefore, the authors has to improve the research problem and scientific depth of present study in order to consider it further for publication this journal. Authors must conduct an in-depth literature review of auxetic structures to understand what has been done on this subject and improve present study accordingly. E.g research of Ashby et al, Gibon et al, Jeng et al. Albert To et al. etc. Also, go through the recently published review articles on lattice structures to find out the research gap for improving present study.

Author Response

We appreciate the opportunity to review our article, and we are convinced that the novelty and contribution to the scientific community of metamaterials will be well received. According to the reviewer's comment, we agree that the objective and novelty of the study was not well defined in the previous version of the manuscript not given in the abstract, which has now been corrected in this new version.  In general, 2D reentrant structures are symmetric, but in this case as a new feature, a design parameter alpha is introduced in order to produce asymmetry in the 2D structure. To validate the proposal, the performed quasi-static compression experiments show that the structure responds with different elastic properties in its three orthogonal directions. We are sure that these results are interesting to several researchers, so we highly recommend it to be considered for publication. Since this field of research is current and has grown very rapidly in recent years, many potential applications of this new proposed method are key for the design of tailored auxetic structures. The results of this research allow the design of a structure that requires different mechanical behavior depending on the direction of work, from the simple unit cell analysis. Furthermore, the results show that this asymmetric cell could transmit to the structure a different stiffness behavior in three working directions, which would allow, for example, its application in the fabrication of an ankle prosthesis.

Reviewer 4 Report

There is a problem with this article. When I use the button “Download Manuscript” I get the file “materials-1452048-peer-review-v2.pdf”. When I use “Show/hide old version” and “Download Manuscript v1” I get the file “materials-1452048-peer-review-v1.pdf”. The surprising thing is that BOTH versions contain errors:
(a) v1-version has 47 references and mistakes in equations (11) and (12) whereas
(b) in the v2-version the mentioned equations seem to be OK but there are MANY mistakes in 48 references, e.g. the reference [41] K. E. Evans, “Tensile network microstructures exhibiting negative Poisson’s ratio,” J. Phys. D. Appl. Phys., vol. 22, no. 12, pp. 1870–1876, 1989, doi: 10.1088/0022-3727/22/12/011 is cited as [36].
The further text of this review concerns the version “v2”.

In the article “Design and characterization of asymmetric cell structure of auxetic material with directional mechanical response” its authors, Rodrigo Valle, Gonzalo Pincheira, Víctor Tuninetti, Eduardo Fernandez, and Esmeralda Uribe-Lam, discuss elastic properties of a simple three-dimensional (3D) structure. The latter is based on a well-known two-dimensional  (2D) structure which is auxetic, i.e. exhibits negative Poisson’s ratio. Usually the 2D re-entrant structures are symmetric but the authors describe their exercise with asymmetric structures. Samples of the studied 3D structure were manufactured by the authors and tested under quasi-static compression to determine their elastic properties – the Poisson’s ratio and Young’s modulus. The authors show that the proposed structure has different elastic properties in its three orthogonal directions. Although this result, as well as other results presented in this paper, cannot be seen as surprising, the article may be of interest for some researchers and hence worth of publishing. This is because many potential applications have been designed for auxetics. So, the field is topical and quickly growing. However, before this article can be recommended for publication, it requires a major revision. Below are some remarks and suggestions which may be helpful for authors if they decide to re-submit their article.

(A) I think that it would be important for this article if the authors could present in the introduction at least a few mechanisms of auxeticity. The paper would be much stronger if the authors could also discuss, in more detail, auxetic (meta)materials, their structures, properties, models and applications. In particular:

  1. The first negative Poisson’s ratio (NPR) material were foams manufactured by Lakes, see [R. Lakes, FOAM STRUCTURES WITH A NEGATIVE POISSONS RATIO, SCIENCE, Volume: 235, Pages: 1038-1040, Published: FEB 27 1987].
  2. The first molecular model of isotropic auxetic was studied by Monte Carlo simulations in [K. W. Wojciechowski, CONSTANT THERMODYNAMIC TENSION MONTE-CARLO STUDIES OF ELASTIC PROPERTIES OF A TWO-DIMENSIONAL SYSTEM OF HARD CYCLIC HEXAMERS, MOLECULAR PHYSICS, Volume: 61, Pages: 1247-1258, Published: ‏ AUG 10 1987] and solved analytically at zero temperature in [K. W. Wojciechowski, TWO-DIMENSIONAL ISOTROPIC SYSTEM WITH A NEGATIVE POISSON RATIO, PHYSICS LETTERS A, Volume: 137, Pages: 60-64, Published: MAY 1 1989].
  3. 2D reentrant structure was mentioned in [L.J. Gibson, M.F. Ashby, Cellular Solids: Structure and Properties; Pergamon Press: Oxford, UK, 1988].
  4. 3D re-entrant structure was presented in [R.F. ALMGREN, AN ISOTROPIC 3-DIMENSIONAL STRUCTURE WITH POISSON RATIO=-1; JOURNAL OF ELASTICITY, Volume 15, Pages: 427-430, Published: 1985].
  5. Re-entrant structures constitute the simplest examples of auxetic (meta)materials. Recently, it has been shown,  however, that not only structure but also material is crucial and one can obtain re-entrant structures which are not auxetic. This issue is of relevance for this paper and should be discussed.
  6. Rotating square model was presented in [J. N. Grima, et al., Auxetic behavior from rotating squares, JOURNAL OF MATERIALS SCIENCE LETTERS, Volume: 19, Pages: 1563-1565, Published: ‏ SEP 2000].
  7. Modern auxetic structures can be made of natural materials, e.g. wood-based sandwich panels and sandwich beams were recently proposed in the literature. As ecological materials become more and more important in the present technology and everyday life, it would be worth to discuss that aspect of auxetics.

(B) The parameter \alpha should be defined not only in Fig.1 but also explicitly, using a formula. Moreover, this “distance factor” should be represented in the article by the same symbol – a Greek letter, not a proportionality symbol.

(C) Results are presented in a bit chaotic way. For pedagogical reasons, before discussing 3D structures it would be very important to discuss (in a separate paragraph and in detail) 2D asymmetric model and present the results (both analytic and experimental) for the same parameters as it is done for 3D cases.

(D) I think that instead of presenting the Poisson’s ratio in the form of 3D-plot (Fig.3) it would be more clear to present it in 2D-plots as function of the re-entrant angle for some values of the parameter \alpha.

(E) To study the size dependence of the experimental results, what is important as theoretical results are obtained for “infinite” system, samples of different sizes should be prepared and studied. The results should be properly plotted to make possible an extrapolation to the analytic model.

(F) Results obtained for Timoshenko beams were compared with finite element method simulations in [A. A. Pozniak, et al., Computer simulations of auxetic foams in two dimensions, SMART MATERIALS AND STRUCTURES, Volume: 22, Article Number: 084009, Published: AUG 2013, DOI: 10.1088/0964-1726/22/8/084009]. I think that it would be interesting to perform some finite element method simulations (the authors in conclusions that such computations are planned to be done in future) as in such a case there is no problem with the size dependence of the results – one can use periodic boundary conditions.

(G) Careful reading and language polishing of the manuscript by a native speaker would be useful to eliminate some imperfections like “There are analytical analysis studies” (page 2), etc.

(H) There are many mistakes in the references, which should be checked and corrected. These mistakes make the reviewing process very difficult.

In conclusion, the present version of the manuscript cannot be published. The authors should revise the paper and it should be reviewed once again.

Author Response

We appreciate the opportunity offered for improving our manuscript and resubmit it to the journal Materials for a second round of revisions. We have addressed all the points raised by the reviewers 1, 3, 4 and 5. Regarding the comments of reviewer 1, we have improved all the recommendations in order to be considered for publication. Regarding the comment of reviewer 4, authors have tried to answer the reviewer's questions in the best way possible with specific information. Additionally, we have included all the recommendations.

Reviewer 5 Report

This work is devoted to theoretical and experimental investigations of 3D auxetic structure based on a known planar configuration. The novelty of this work consists in consideration of the asymmetric cell structure, whereas the similar symmetric cell structure was analyzed yet in 1996 [Masters I G and Evans K E 1996 Models for the elastic deformation of honeycombs Compos. Struct. 35 403–22] Here, the authors introduced an additional parameter admitting variation of elastic properties of the structure along three orthogonal directions. To model the mechanical behavior of the cell in its three orthogonal directions, the authors used Timoshenko beam theory. Experimental samples were constructed using a 3D printer Stratasys uPrint SE, equipped with Fused Deposition Modeling (FDM) technology. It was shown that, in general, the Poisson's ratio curves as a function of the design parameter ∝ show high accuracy with the experimental results. The theoretical model achieves capture the dependence of the Poisson's ratio with the parameter ∝. However, in some cases the disagreement can be as high as 20%, such as structures designed with a 50° re-entrant angle. The authors have identified several possible reasons for such disagreement.

I recommend publication of this paper after a revision concerning English language and text. For example, I would advise to correct the text fragment from Section 3.1 in order to avoid duplicates:

  "But it should be noted that the theoretical analysis is based on the     assumptions that the structure has an infinite pattern of cells in the three  orthogonal directions and considering only the bending deformation through Timoshenko's theory. Furthermore, the analysis only considers the bending deformation through Timoshenko's theory. While the measurements on the samples that have a finite number of cells and the samples experience multiple deformations such as axial and shear."

by the following:

  "But it should be noted that the theoretical analysis is based on the assumptions that the structure has an infinite pattern of cells in the three orthogonal directions. Furthermore, the analysis considers only the bending deformation within the scope of Timoshenko's theory, while the measurements on the samples that have a finite number of cells and the samples experience multiple deformations such as axial and shear."

Author Response

We appreciate the opportunity offered for improving our manuscript and resubmit it to the journal Materials for a second round of revisions. We have addressed all the points raised by the reviewers 1, 3, 4 and 5. Regarding the reviewer comment 5, the authors have tried to respond to the suggestions made, additionally the manuscript has been revised by a native speaker to improve its translation

Round 2

Reviewer 1 Report

Can be accepted in present form

Author Response

We appreciate the opportunity to recommend our article for publication.

Reviewer 3 Report

Authors have improved and addressed the comments and concerns accordingly. However needs to go through following papers to understand 3d reentrant structures and also include them in references.

https://journals.sagepub.com/doi/pdf/10.1177/0021998318764021

https://link.springer.com/referenceworkentry/10.1007%2F978-981-10-6884-3_25

In addition, authors also needs to include following recently published review article. https://link.springer.com/article/10.1007%2Fs00170-019-04085-3

Author Response

Dr. Gonzalo Pincheira

University of Talca

Dpt. Industrial Technologies

Camino Los Niches. CURICÓ (CHILE)

January 13th, 2022

Dear Ms. Joey Shao,

We appreciate the opportunity offered for improving our manuscript and resubmit it to the journal Materials for a second round of revisions. We have addressed all the points raised by the reviewers 3 and 4. Regarding the reviewer's comment, 3 the authors have included the suggested references to the state of the art of the article.

To conclude, all specific reviewers' comments were considered and included in the review. Please note that modified and inserted texts are clearly highlighted, because we have used the “track of changes” option in Microsoft Word (materials-1452048_track_changes). Additionally, we have added a final version of the work with all the changes accepted (materials-1452048_final). Finally, together with this letter, you will find two letters addressed to reviewers addressing their comments. We hope the changes we made in our paper will make it suitable to be published in your journal.

Thank you very much for your consideration.

Yours sincerely,

The authors

DETAILED RESPONSE TO REVIEWERS

Journal: Materials

Manuscript ID:  materials-1452048

Title: Design and characterization of asymmetric cell structure of auxetic material with directional mechanical response

Reviewer  #3

Authors have improved and addressed the comments and concerns accordingly. However needs to go through following papers to understand 3d reentrant structures and also include them in references.

  • https://journals.sagepub.com/doi/pdf/10.1177/0021998318764021
  • https://link.springer.com/referenceworkentry/10.1007%2F978-981-10-6884-3_25

In addition, authors also needs to include following recently published review article.

  • https://link.springer.com/article/10.1007%2Fs00170-019-04085-3

>>Authors: We appreciate suggestions to improve the state of the art of our article. We have included the indicated references in the Introduction, as shown in the following text: [new text in red color]

For example, higher mechanical strength and rigidity are obtained with the increment of relative density. However, tensile-dominated cell designs may experience a higher Young’s modulus than bending-dominated designs [30], [31] while maintaining constant density. In [32] an exhaustive review of the various morphologies of cell structures is presented along with the superior properties that can be obtained, as well as their applications and challenges. A type of cell design that has attracted much attention is auxetic structures. Unlike conventional designs, they exhibit a negative Poisson's ratio, i.e., they expand laterally when stressed and contract laterally when compressed. In [33] a review of the deformation mechanisms that allow auxetic behavior and the improved characteristics produced by a negative poisson's ratio is provided. This auxetic effect provides remarkable mechanical properties, such as low specific density, high shear modulus, and higher energy absorption capacity [34]–[42]; properties highly required for the design of lightweight structures. Several design approaches for auxetic structures have been reported in the literature, from the pioneering analytical study shown by Almgren [43] of an isotropic auxetic structure with a Poisson's ratio , through the work of Lakes in [44] where a foam structure with negative Poisson's ratio is fabricated, to the works of Wojciechowski in [45], [46] where a two-dimensional auxetic structure is studied through constant thermodynamic stress Monte Carlo simulations in order to determine its elastic properties. Subsequently, Gibson in [47] and Evans in [48] present an auxetic structure with reentrant struts whose microstructure is modeled to exhibit transverse expansion under longitudinal loading. Finally, current studies reported where carbon fiber reinforced polymer additive manufacturing technology has been used using an interlocking assembly method in the fabrication of three-dimensional auxetic structures [49]. In addition, current research provides evidence and potential applications for the wood industry, specifically in furniture manufacturing, because plywood and bending processes consume significant time and resources, which could be considerably improved by implementing auxetic structures.

Reviewer 4 Report

I guess that the authors have done what they were able to do. The manuscript has been amended. However it still requires a minor revision as suggested below.

1. In the abstract the authors write about "a novel design parameter producing asymmetry". I think that the word "novel" is too strong for this paper and might be removed.

2. Regarding the point A5 of my review, which was:

  1. Re-entrant structures constitute the simplest examples of auxetic (meta)materials. Recently, it has been shown, however, that not only structure but also material is crucial and one can obtain re-entrant structures which are not auxetic. This issue is of relevance for this paper and should be discussed.

the authors may consult the below papers:

(i) anisotropic re-entrant structures of highly positive Poisson’s ratio [M. Bilski, et al., Extreme Poisson's Ratios of Honeycomb, Re-Entrant, and Zig-Zag Crystals of Binary Hard Discs, SYMMETRY-BASEL, Volume: 13, Article Number: 1127, Published:‏ JUL 2021],

(ii) isotropic re-entrant structures of highly positive Poisson’s ratio [M. Bilski, et al., Extremely Non-Auxetic Behavior of a Typical Auxetic Microstructure Due to Its Material Properties, MATERIALS, Volume: 14, Article Number: 7837, Published:‏ DEC 2021].

Author Response

Dr. Gonzalo Pincheira

University of Talca

Dpt. Industrial Technologies

Camino Los Niches. CURICÓ (CHILE)

January 13th, 2022

Dear Ms. Joey Shao,

We appreciate the opportunity offered for improving our manuscript and resubmit it to the journal Materials for a second round of revisions. We have addressed all the points raised by the reviewers 3 and 4. Regarding the reviewer's comment, 4 the authors have included the references suggested in the discussion of the experimental results of the article.

To conclude, all specific reviewers' comments were considered and included in the review. Please note that modified and inserted texts are clearly highlighted, because we have used the “track of changes” option in Microsoft Word (materials-1452048_track_changes). Additionally, we have added a final version of the work with all the changes accepted (materials-1452048_final). Finally, together with this letter, you will find two letters addressed to reviewers addressing their comments. We hope the changes we made in our paper will make it suitable to be published in your journal.

Thank you very much for your consideration.

Yours sincerely,

The authors

DETAILED RESPONSE TO REVIEWERS

Journal: Materials

Manuscript ID:  materials-1452048

Title: Design and characterization of asymmetric cell structure of auxetic material with directional mechanical response

Reviewer #4

  1. In the abstract the authors write about "a novel design parameter producing asymmetry". I think that the word "novel" is too strong for this paper and might be removed.

>>Authors: We appreciate the suggestion to improve our article. We have removed the word "Novel".

Abstract

A three-dimensional auxetic structure based on a known planar configuration including a design parameter producing asymmetry is proposed in this study. The auxetic cell is designed by topology analysis using classical Timoshenko beam theory in order to obtain the required orthotropic elastic properties. Samples of the structure are fabricated using the ABSplus fused filament technique and subsequently tested under quasi-static compression to statistically determine the poisson’s ratio and Young’s modulus. The experimental results show good agreement with the topological analysis and reveal that the proposed structure can adequately provide different elastic properties in its three orthogonal directions. In addition, three point bending tests were carried out to determine the mechanical behavior of this cellular structure. The results show that this auxetic cell influences the macrostructure to exhibit different stiffness behavior in three working directions.

  1. Regarding the point A5 of my review, which was:

Re-entrant structures constitute the simplest examples of auxetic (meta)materials. Recently, it has been shown, however, that not only structure but also material is crucial and one can obtain re-entrant structures which are not auxetic. This issue is of relevance for this paper and should be discussed.

the authors may consult the below papers:

(i) anisotropic re-entrant structures of highly positive Poisson’s ratio [M. Bilski, et al., Extreme Poisson's Ratios of Honeycomb, Re-Entrant, and Zig-Zag Crystals of Binary Hard Discs, SYMMETRY-BASEL, Volume: 13, Article Number: 1127, Published:‏ JUL 2021],

(ii) isotropic re-entrant structures of highly positive Poisson’s ratio [M. Bilski, et al., Extremely Non-Auxetic Behavior of a Typical Auxetic Microstructure Due to Its Material Properties, MATERIALS, Volume: 14, Article Number: 7837, Published:‏ DEC 2021].

>>Authors: We appreciate the contribution of these references in order to improve our article. Therefore, we have included in the discussion these new references in the following text:  [new text in red color]

To determine the elastic behavior of this cell structure, nine combinations are achieved for the Poisson's ratio: three groups for different values ​​of the re-entrant angle  and three groups for the new design parameter . The theoretical and experimental results for Poisson’s ratio are shown in Figure 9. In general, the Poisson's ratio curves as a function of the new design parameter  show high accuracy with the experimental results. The theoretical model achieves capture the dependence of the Poisson's ratio with the parameter . As the structure experiences a homogeneous deformation under compression in the x direction, the Poisson's ratio does not vary as a function of , but slightly decreases as a function of the re-entrant angle. On the other hand, due to the asymmetry of cell 2 shown in Figure 1 (b), the Poisson's ratio  becomes more negative the smaller the value of  and it decreases slightly the smaller the re-entrant angle. Which helps to validate the proposed design. However, in some cases the disagreement can be as high as 20%, such as structures designed with a 50° re-entrant angle. But it should be noted that the theoretical analysis is based on the assumptions that the structure has an infinite pattern of cells in the three orthogonal directions and considering only the bending deformation through Timoshenko's theory. Furthermore, the analysis only considers the bending deformation through Timoshenko's theory. While the measurements on the samples that have a finite number of cells and the samples experience multiple deformations such as axial and shear. On the other hand, the application of the simplified model becomes more complex to apply to this cell because of the asymmetry of cell 2 shown in Figure 1 (b). Another possible cause of the differences between the theoretical predictions and the experiments is the variability in the size of the strut and the quality of the surface produced by the manufacturing process, particularly considering that structures of this type with re-entrant struts inevitably have defects such as the well-known step effect produced by additive processes [60]. On the other hand, the Poisson's ratio obtained through the Euler-Bernoulli approximation only depends on the geometry of the structure and does not consider the properties of the material. According to research carried out in [61], [62], auxetic structures that contradict the theoretical behavior are studied by experiencing a positive Poisson's ratio. In this way, these results show that the properties of these metamaterials not only depend on their microstructure and that the manufacturing material can be crucial. Therefore, it is a challenge to model with great precision the transverse displacement of this type of structures under compressive loads.
